# Solution structure of human myeloid-derived growth factor suggests a conserved function in the endoplasmic reticulum

Valeriu Bortnov [1], Marco Tonelli[2,3], Woonghee Lee [2,3], Ziqing Lin [4,5], Douglas S. Annis [1], Omar N. Demerdash[6], Alex Bateman[7], Julie C. Mitchell [6], Ying Ge [4,5], John L. Markley[2,3] & Deane F. Mosher[1,8]*

Human myeloid-derived growth factor (hMYDGF) is a 142-residue protein with a C-terminal endoplasmic reticulum (ER) retention sequence (ERS). Extracellular MYDGF mediates cardiac repair in mice after anoxic injury. Although homologs of hMYDGF are found in eukaryotes as distant as protozoans, its structure and function are unknown. Here we present the NMR solution structure of hMYDGF, which consists of a short α-helix and ten β-strands distributed in three β-sheets. Conserved residues map to the unstructured ERS, loops on the face opposite the ERS, and the surface of a cavity underneath the conserved loops. The only protein or portion of a protein known to have a similar fold is the base domain of VNN1. We suggest, in analogy to the tethering of the VNN1 nitrilase domain to the plasma membrane via its base domain, that MYDGF complexed to the KDEL receptor binds cargo via its conserved residues for transport to the ER.

[1] Department of Biomolecular Chemistry, University of Wisconsin-Madison, Madison, WI 53706, USA. [2] Department of Biochemistry, University of Wisconsin-Madison, Madison, WI 53706, USA. [3] National Magnetic Resonance Facility at Madison, University of Wisconsin-Madison, Madison, WI 53706, USA. [4] Departments of Cell and Regenerative Biology and Chemistry, University of Wisconsin-Madison, Madison, WI 53706, USA. [5] Human Proteomics Program, University of Wisconsin-Madison, Madison, WI 53706, USA. [6] Biosciences Division, Oak Ridge National Laboratory, Oak Ridge, TN 37831, USA. [7] European Molecular Biology Laboratory, European Bioinformatics Institute (EMBL-EBI), Wellcome Genome Campus, Hinxton CB10 1SD, UK. [8] Department of Medicine, University of Wisconsin-Madison, Madison, WI 53706, USA. *email: dfmosher@wisc.edu

Human myeloid-derived growth factor (hMYDGF) is a member of the widely distributed MYDGF family of proteins found in organisms as distant as protozoans[1]. In humans, MYDGF is abundant in nearly 200 different tissues, fluids, and cell lines as compiled by Proteomics DB, a repository of quantitative proteomics data[2]. hMYDGF comprises a 31-residue signal sequence followed by a 142-residue mature protein ending in C-terminal residues RTEL. Experiments appending candidate endoplasmic reticulum (ER) retention sequences (ERS) to the C-terminus of a model secreted protein for expression in HeLa cells demonstrated that RTEL binds to human KDEL receptors KDELR1, KDELR2, and (to a lesser extent) KDELR3 (ref. [3]). KDELRs retain target proteins in the ER by engaging a C-terminal ERS at the slightly acidic pH of the Golgi (6.0–6.7) and dissociating under the more neutral pH conditions of the ER (7.2) after retrograde transport[4–6]. Removal of the C-terminal Glu-Leu residues from hMYDGF exogenously expressed in HEK293 cells demonstrated that absence or presence of an intact ERS determines, in a nearly all-or-none fashion, whether hMYDGF is retained in ER or secreted[1].

The name "myeloid-derived growth factor" was assigned when MYDGF was identified as a protein secreted from monocytes/macrophages that promotes tissue repair in a murine model of myocardial infarction[7]. MYDGF-knockout mice, which lacked a developmental phenotype, developed larger infarct scars and impaired angiogenesis compared to control mice after cardiac ischemia followed by reperfusion[7]. Delivery of recombinant mouse MYDGF also ameliorated effects of cardiac ischemia and improved survival of mice in which MYDGF was not knocked out[7]. Human plasma has a median hMYDGF concentration of 3.3 ng/mL (0.2 nM) as analyzed by multiple reaction monitoring-mass spectrometry, with increased levels in patients with acute myocardial infarctions[8]. Endogenous MYDGF, other resident ER proteins, and a reporter construct containing the last 7 residues of hMYDGF have been shown to be released from SH-SY5Y human neuroblastoma cells in response to ER calcium depletion by thapsigargin[9]. The same result was obtained with cells exposed to oxygen-glucose deprivation[9], an in vitro model of ischemia tied to depleted intracellular calcium stores[10]. Thus, the literature indicates that although MYDGF is predominantly retained in the ER as a result of engaging KDELRs through its ERS, it is secreted upon cellular stress to act as a paracrine/autocrine survival factor with therapeutic potential.

MYDGF homologs are grouped into the Uncharacterized Protein Family (UPF) 0556 in the Pfam database (Pfam: PF10572 (http://pfam.xfam.org/family/PF10572)), which is annotated on the basis of sequence analysis as having unknown structure and no similarity to any other protein family. To fill the gap in knowledge regarding the structure of a protein expressed widely in nature and guide future functional studies of MYDGF, we now report the solution structure of hMYDGF determined at pH 6 by protein NMR. We demonstrate that the face of hMYDGF predicted to interact with KDELRs is perturbed by changes in pH, whereas residues conserved in MYDGF homologs are clustered on the opposite face. A search for structural folds similar to hMYDGF identified the base domain of the vanin family of pantetheinases, which tethers the vanin nitrilase domain to the outer plasma membrane. We suggest, in analogy to vanins, that MYDGF complexed to KDELRs binds cargo via its conserved residues for transport to the ER.

## Results

**Production and characterization of recombinant hMYDGF.** A gene coding for hMYDGF lacking the signal peptide residues (mature hMYDGF: V32-L173; UniProt Q969H8 (https://www.uniprot.org/uniprot/Q969H8)) with a thrombin-cleavable N-terminal 6xHis-tag was expressed in *Escherichia coli* cells grown in medium containing [U-$^{13}$C]-glucose and $^{15}$N-ammonia as, respectively, the sole sources of carbon and nitrogen. The protein was extracted in 8 M urea after cell lysis, purified by immobilized metal affinity chromatography (IMAC), renatured by dialysis against dilute acetic acid (pH 3.7) and then against Tris-buffered saline (TBS; pH 8.5), treated with thrombin to remove the N-terminal 6xHis-tag, and purified from the cleavage mix by size exclusion chromatography. The final product comprised the mature hMYDGF with the sequence GSKGT introduced at the N-terminus as a cloning artifact.

To evaluate whether hMYDGF folded and purified by this method has the same structure as hMYDGF processed through the ER, we compared the bacterially expressed protein to hMYDGF secreted from High Five insect cells. Cells were infected with recombinant baculovirus encoding the mature hMYDGF sequence flanked by an N-terminal gp67 signal sequence that targets the protein to the ER and a C-terminal 6xHis-tag that masks the ERS, enabling the construct to be secreted and subsequently isolated and purified from culture medium by IMAC. We performed top-down mass spectrometry (MS)[11] to confirm the sequences of the proteins and to determine the oxidation state of the two conserved cysteines (C63, C92). The experimental molecular masses of the tagged insect cell-derived and untagged bacteria-derived hMYDGF matched the masses calculated for the proteins with oxidized cysteines (Supplementary Fig. 1a, c). Tandem MS (MS/MS) of the insect cell-derived construct by collision-induced dissociation (CID) and the bacteria-derived construct by CID or electron-transfer dissociation (ETD), which produce backbone cleavages in the protein, did not yield fragmentation between C63 and C92 (Supplementary Fig. 1b, d), confirming that hMYDGF forms a disulfide bond in both expression systems. As a second comparison, C-terminus tagged insect cell-derived hMYDGF and N-terminus tagged bacteria-derived hMYDGF were examined by far UV circular dichroism (CD) spectroscopy (Supplementary Fig. 1e, Supplementary Table 1). The two spectra overlaid closely, with both exhibiting small negative peaks centered around 231 nm and 218 nm and a large positive peak near 202 nm. The secondary structure content predicted by the Beta Structure Selection (BeStSel) web server[12] on the basis of the CD spectra of the two constructs also closely matched: both with ~40% antiparallel β-strand, 2% α-helix, and ~58% irregular (Supplementary Table 1). Finally, we measured the intrinsic fluorescence of the two hMYDGF tryptophan residues (W77, W95) (Supplementary Fig. 1f). Excitation at 295 nm produced single emission peaks for the tagged insect cell-derived protein and untagged bacteria-derived protein at 338.5 nm and 338.0 nm, respectively, and nearly identical peak contours. We conclude that the refolding protocol for bacterially expressed hMYDGF yielded a product experimentally equivalent to hMYDGF secreted from a eukaryotic cell. These experiments, therefore, justify use of the recombinant protein isotopically labeled by *E. coli* to determine the native structure of hMYDGF.

**Solution structure of hMYDGF solved by protein NMR.** The structure of [U-$^{13}$C, $^{15}$N]-hMYDGF in solution at pH 6 was determined by protein nuclear magnetic resonance (NMR) spectroscopy. This pH is at the lower end of the Golgi pH range at which KDELRs efficiently engage ERS-containing proteins[4,13]. In brief, two-dimensional (2D) and three-dimensional (3D) NMR spectra were used to assign the $^{1}$H, $^{13}$C, and $^{15}$N resonances from the backbone and sidechains of recombinant hMYDGF; the chemical shift assignment completeness was 91% for all atoms.

**Table 1 NMR and refinement statistics for the solution structure of hMYDGF.**

| NMR constraints and structure statistics | hMYDGF |
|---|---|
| Distance constraints | |
| Total NOE | 3703 |
| Short-range ($\lvert i - j \rvert \leq 1$) | 2932 |
| Medium-range ($1 < \lvert i - j \rvert \leq 5$) | 151 |
| Long-range ($\lvert i - j \rvert > 5$) | 620 |
| Hydrogen bonds | 28 |
| Dihedral angle restraints | |
| Total | 239 |
| $\phi$ | 117 |
| $\psi$ | 122 |
| Structure statistics[a] | |
| Violations | |
| Distance constraints (>0.5 Å) | 0 |
| Dihedral angle constraints (>5°) | 0 |
| Van der Waals (>0.2 Å) | 0 |
| Average pairwise RMSD[b] (Å) | |
| Heavy | 1.30 ± 0.09 |
| Backbone | 0.72 ± 0.07 |
| Xplor-NIH pseudopotential energy (kJ mol$^{-1}$) | 5481 |
| MOLPROBITY mean score/clash score | 2.10/11.21 |
| MOLPROBITY Ramachandran plot summary[b] (%) | |
| Favored regions | 95.7 |
| Allowed regions | 4.3 |
| Disallowed regions | 0.0 |

[a]Structure statistics were calculated using the 20 lowest pseudo-potential energy conformers out of the 100 total calculated conformers. The average pairwise RMSD was calculated against the lowest-energy conformer. [b]RMSD and Ramachandran statistics were obtained using ordered hMYDGF residues P35-A126 and D133-A168 as defined by CYRANGE

The resonance assignments and 3D NOESY spectra were utilized for Xplor-NIH-based structure calculation through the PONDEROSA-C/S suite, in an iterative process of structure calculation and constraint validation. This method calculated the top 100 most energetically stable hMYDGF conformers representing the 3D structure, from which the NMR statistics of the top 20 are summarized in Table 1. The final structure calculation was based on 3703 distance (620 long-range, 151 medium-range, 2932 short-range), 239 dihedral angle, and 28 hydrogen bond constraints with no constraint violations among the top 20 conformers. Backbone phi/psi angles in these structures were in favored regions of the Ramachandran plot for 96% of all ordered residues as assessed by MOLPROBITY, with none falling in disallowed regions. NMR data were deposited in the BioMagResBank[14] (BMRB 30584 (http://www.bmrb.wisc.edu/data_library/summary/?bmrbId=30584)), and the structural coordinates and restraints were deposited in the Protein Data Bank[15,16] (PDB 6O6W (https://www.rcsb.org/structure/6O6W)).

As shown for the most energetically stable conformer (Fig. 1a), the global fold of hMYDGF comprises three antiparallel β-sheets (ten β-strands: β1–β10) and a single α-helical turn (α1). The largest β-sheet (β1, β4, β5, β10, and β7; red) is linked to a smaller β-sheet (β2, β3, and β6; orange) by the disulfide bridge (stick representation between β3 and β5) to form a β-sandwich. The β-sandwich is capped at the β6/β7 edge by the α-helical turn (blue) and the third β-sheet (β8 and β9; green). The two terminal ends of hMYDGF are on the same face of the protein, with the five N-terminal residues having been introduced as a result of cloning (black) and the four C-terminal residues (RTEL) comprising the ERS (yellow). A schematic of β-strand connectivity is depicted in Fig. 1b, color-coded by β-sheet with the disulfide bridge represented by a dotted line. Positively charged (blue), negatively

charged (red), and uncharged (white) patches are scattered over the protein surface (Fig. 1c).

The 20 most energetically stable hMYDGF conformers (Fig. 1d) align with a root-mean-square deviation (RMSD) of 1.3 Å for all heavy atoms and 0.7 Å for backbone heavy atoms of ordered residues (Table 1). The regions with the highest degree of flexibility are located at the N-terminus, C-terminus, and the elongated loop between β7 and α1, which projects outward from the globule. These and other regions of hMYDGF lacking secondary structure align well with $^{15}$N relaxation data, which showed longer $T_2$ and smaller heteronuclear nuclear Overhauser effect (NOE) values when compared to the more structured sections of the backbone (Supplementary Fig. 2). The average relaxation times calculated from amides in secondary structure, 763.6 ms for $T_1$ and 78.6 ms for $T_2$, were used to estimate a rotational correlation time ($\tau_c$) of 9.4 ns for hMYDGF (Equation 2 in Rossi et al.[17]). This value corresponds to a molecular mass of 15.5 kDa[17], which is close to the recombinant hMYDGF mass of 16.25 kDa and confirms the protein is monomeric in solution under the tested conditions.

The secondary structure composition of the hMYDGF NMR structure is 48% antiparallel β-strand, 2% α-helix, and 50% irregular. This aligns well with the secondary structure predicted by BeStSel[12] of 45% antiparallel β-strand, 0% α-helix, and 54% irregular calculated from CD spectra of hMYDGF in buffers ranging from pH 4.0 to 7.5 (Fig. 2a, Supplementary Table 1). The two tryptophan residues, while in close proximity, are on opposite sides of the largest β-sheet: W77 oriented toward the core on β4 and W95 toward the solvent on β5 (Supplementary Fig. 3a). However, only a single tryptophan fluorescence emission peak at 338 nm was present after excitation at 295 nm (Supplementary Fig. 3b). The peak was not perturbed by addition of 10 mM dithiothreitol (DTT), whereas addition of 6 M guanidine without or with DTT resulted in a single peak red-shifted to ~357 nm and increased fluorescence intensity (Supplementary Fig. 3b). These results indicate that the tryptophan residues share a hydrophobic environment and transition to a hydrophilic environment upon denaturation. Consistent with this interpretation, we did not observe backbone or sidechain amide peaks for either tryptophan in clean solvent exposed amide (SEA) heteronuclear single quantum correlation (HSQC) spectra recorded with mixing times ranging from 10 to 140 ms to allow for hydrogen exchange with bulk water (Supplementary Fig. 3c, d; 140 ms spectrum presented). This result indicates that solvent exchange is relatively slow for both tryptophan residues.

**pH and calcium titration of hMYDGF.** Because hMYDGF may reside in multiple cellular microenvironments including calcium-rich ER (pH 7.2), Golgi (pH 6.0–6.7), and potentially other intracellular compartments of the secretory and endocytic pathways (pH down to 5.5[4]), we determined whether the structure of hMYDGF varies as a function of pH or calcium concentration. The overall fold of hMYDGF as assessed by CD spectroscopy was not sensitive to pH (Fig. 2a, Supplementary Table 1). $^1$H, $^{15}$N HSQC peaks from individual residues of hMYDGF, however, exhibit pH-dependent changes. Figure 2b displays an overlay of $^1$H, $^{15}$N HSQC spectra of hMYDGF at pH conditions ranging pH 5.5–8.0 colored-coded maroon to blue. Inasmuch as all backbone amides except the first two cloning residues have been assigned for hMYDGF at pH 6, we were able to efficiently assign the HSQC spectra recorded at the range of pH values. Peak perturbations for backbone amides varied depending on the residue. For example, the backbone amide peak for E34 remained at the same ppm values for all pH conditions, whereas H49 had the second highest pH shift perturbation (Fig. 2b, expanded view). We

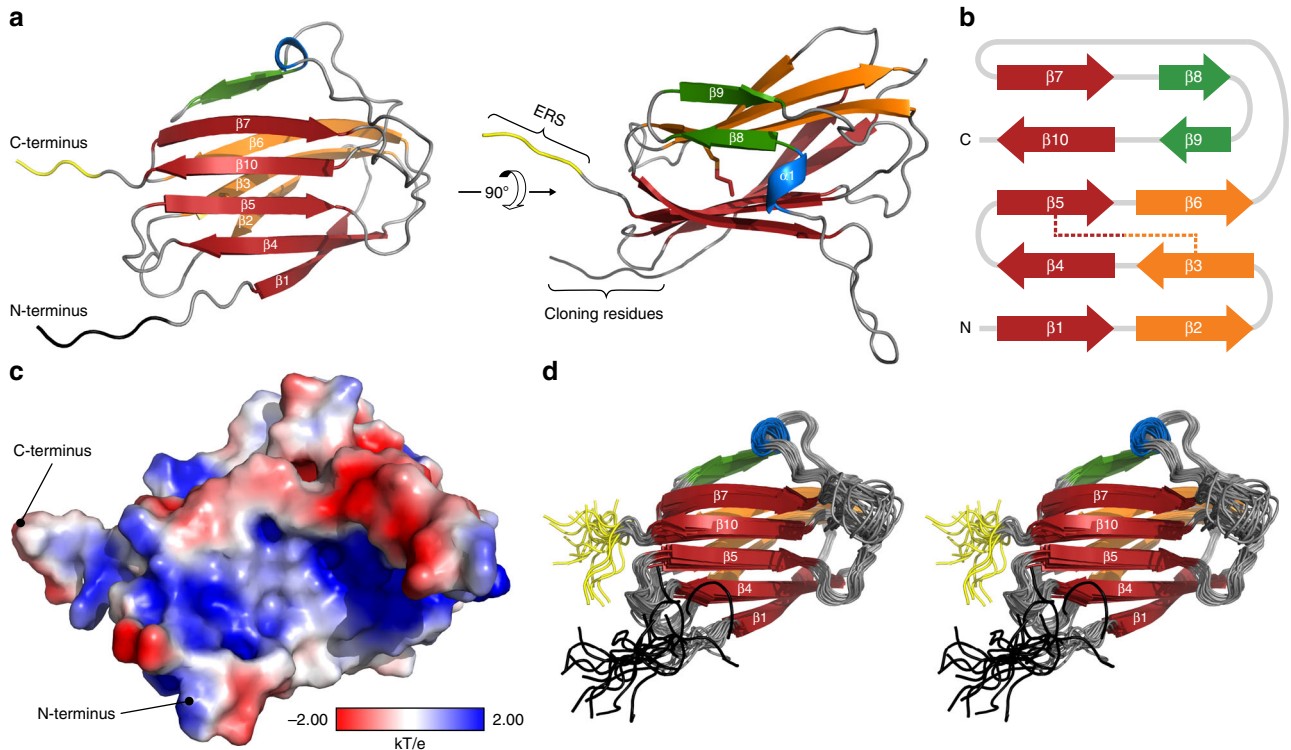

**Fig. 1 Solution structure of hMYDGF solved by protein NMR at pH 6. a** The lowest pseudo-potential energy NMR conformer of hMYDGF. The structure is made up of ten β-strands (β1–β10) that constitute three β-sheets (red, orange, green), a short α-helix (blue), and a disulfide bridge between strands β3 and β5 (right image, stick representation). The first five N-terminal residues were introduced as a result of cloning (black) and the last four C-terminal residues (RTEL) comprise the ERS (yellow). **b** Diagram of the β-strand connectivity colored by β-sheet with the disulfide linkage displayed as a dotted line. **c** Surface charge distribution of hMYDGF (lacking cloning residues) calculated at pH 6 depicting positively charged (blue), negatively charged (red), and uncharged (white) regions scattered across the protein. **d** Overlay of the 20 most energetically stable hMYDGF conformers presented in stereo view. Ordered hMYDGF residues align with an RMSD of 0.72 Å for backbone heavy atoms (Table 1), with the highest degree of flexibility at the N-terminus (black), the C-terminus (yellow), and the loop between β7 and the α-helix (gray).

calculated the peak perturbations ($\Delta\delta_{NH}$) from pH 5.5 to pH 8.0 for each assigned backbone amide and binned them from lowest (gray) to highest (green) ppm difference (Fig. 3a). The residues in the hMYDGF NMR structure were then color-coded to match the $\Delta\delta_{NH}$ bins (Fig. 3b). Since this pH titration crosses the typical $pK_a$ of a histidine imidazole sidechain (~6.0), it is reasonable that areas of the highest peak perturbations surrounded the hMYDGF histidine residues H49, H53, H87, and H89 (Fig. 3a, b; green). The backbone amides of the fifth hMYDGF histidine (H150) and the surrounding residues, however, were not perturbed by variations in pH. The edge of the β-sandwich that includes H53, H87, and H89, and from which the ERS extends, was the most pH-sensitive region (Fig. 3b). Plotting the $\Delta\delta_{NH}$ from each spectrum relative to the pH 5.5 spectrum for the five histidine residues (W95 was included as a pH-independent amide peak) revealed noticeable differences among the titration curves (Fig. 3c). The curve for H150 resembled W95 in that both residue amides were minimally perturbed by pH. The titration curves for the remaining four histidine residues had slightly varying inflection points (most clearly seen when plotted as a percent maximum of $\Delta\delta_{NH}$; Fig. 3c inset), with $pK_a$ values of 6.4 for H49, 6.0 for H87, and 6.8 for H89 calculated from their fitted curves. The $pK_a$ for H53 was estimated to be 5.4 based on an incomplete titration curve. We explored the effect of protonation state on surface charge distribution using the PDB2PQR server[18] and APBS plugin in PyMOL with $pK_a$ values of ionizable groups calculated by PROPKA3.1 (refs. [19,20]) (Fig. 3d). The surface of mature hMYDGF adjacent to the ERS, which includes H53, H87, and H89, is predicted to be predominantly positive at pH 6 and

neutral/negative at pH 7.4 (outlined in Fig. 3d) whereas remaining surface charge distribution is largely unchanged.

We used the HADDOCK protein docking web server[21–23] to dock the most energetically stable hMYDGF NMR conformer onto the crystal structure of chicken KDELR2 (cKDELR2) bound to an ERS-containing peptide (PDB 6I6H (https://www.rcsb.org/structure/6I6H)[6]; peptide was removed for docking calculations) (Supplementary Table 2, Supplementary Fig. 4). The crystal structure, like the NMR structure, was solved at pH 6, and cKDELR2 is 96% identical to human KDELR2 with sidechains of the nine variant residues all facing away from the binding cavity. The 400 complexes calculated by HADDOCK were grouped evenly into two clusters, with hMYDGF less centered along the cKDELR2 cavity and rotated ~130° along the ERS in cluster 2 relative to cluster 1. In the lowest-energy models of both complexes, the C-terminal ERS of hMYDGF (residues RTEL) was bound inside the cKDELR2 cavity and up to 12 additional hMYDGF residues (including H89) were within 5 Å of cKDELR2 that formed interfaces stabilized by multiple polar contacts. Of the close-proximity hMYDGF residues, 12 were the same in cluster 1 (16 total) and cluster 2 (14 total). However, none of the polar contacts with cKDELR2 residues were the same. As highlighted in Fig. 3e (only cluster 1 shown), these interfaces coincide with the location of the pH-sensitive region of hMYDGF depicted in Fig. 3b and d, suggesting that these MYDGF residues have the potential to promote binding to KDELR2 in the Golgi and/or destabilize the complex for MYDGF release in the ER.

Because the ER is the dominant calcium store in the cell with concentrations reaching millimolar[24] and depletion of ER

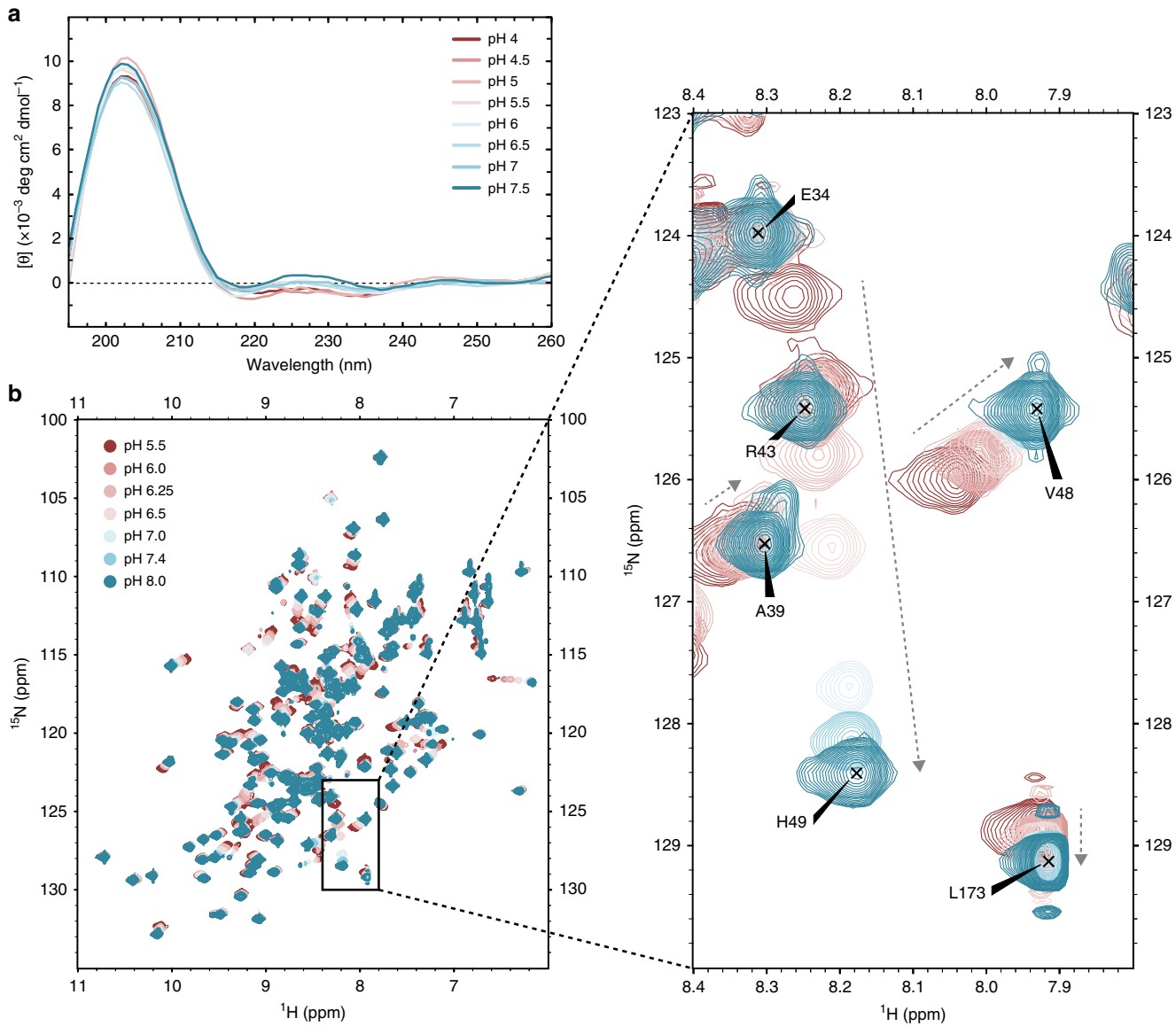

**Fig. 2 Effects of pH on hMYDGF. a** Smoothed CD spectra of hMYDGF in buffer ranging from pH 4.0 (maroon) to pH 7.5 (blue). Spectra were baseline subtracted from their corresponding buffer spectra. Source data for each spectra are provided as a Source Data file. **b** $^1$H, $^{15}$N HSQC spectra of hMYDGF in buffers ranging from pH 5.5 (maroon) to pH 8.0 (blue). Full spectra (left) and a representative expanded view (right) display a range of peak shifts from no movement (e.g., E34), to highly pH-sensitive peaks (e.g., H49). Arrows represent direction of peak perturbation as a result of increasing pH. Assignments shown are for peaks at pH 8.0.

calcium results in secretion of ERS-containing proteins including hMYDGF[9], we determined whether the presence of calcium impacts hMYDGF structure. In contrast to the differences found as a function of pH, the $^1$H, $^{15}$N HSQC spectrum of hMYDGF in the presence of up to 1 mM calcium (fourfold molar excess) was unperturbed relative to hMYDGF in the absence of calcium (Supplementary Fig. 5).

**MYDGF sequence similarity mapped onto the hMYDGF structure**. We used the ConSurf web server[25] to map conserved residues on the structure of hMYDGF. With mature hMYDGF as the input protein sequence, the HMMER homolog search algorithm in ConSurf identified 87 non-redundant MYDGF homologous sequences from the UniRef90 protein database. A phylogenetic tree of these 87 sequences based on MAFFT multiple sequence alignment is presented in Fig. 4a, with UniProtKB/UniParc protein identifiers listed at each branch. The tree is

color-coded based on the MYDGF homolog's phylum and class (Fig. 4a legend), which spans across the animal kingdom (hMYDGF is starred) and also includes slime molds and protozoans. A multiple sequence alignment of one MYDGF homolog from each class (hMYDGF representing Mammalia) is shown in Fig. 4b with the consensus logo and hMYDGF secondary structure underneath the sequences. The 25 positions marked with an asterisk have ≥85% sequence identity across the 236 UniRef90 MYDGF sequences found by Basic Local Alignment Search Tool (BLAST), with the 12 asterisks in black having ≥90% identity. Only ten of the 25 most conserved residues map to regions with secondary structure. The remaining 15 residues map to the ERS and loops that link β1/β2, β3/β4, β5/β6, and β9/β10 strands.

Each residue of the 87 identified homologous sequences was assigned to a conservation level bin by ConSurf; these are color-coded from most variable (orange) to most conserved (blue) and mapped onto the most energetically stable NMR structure of

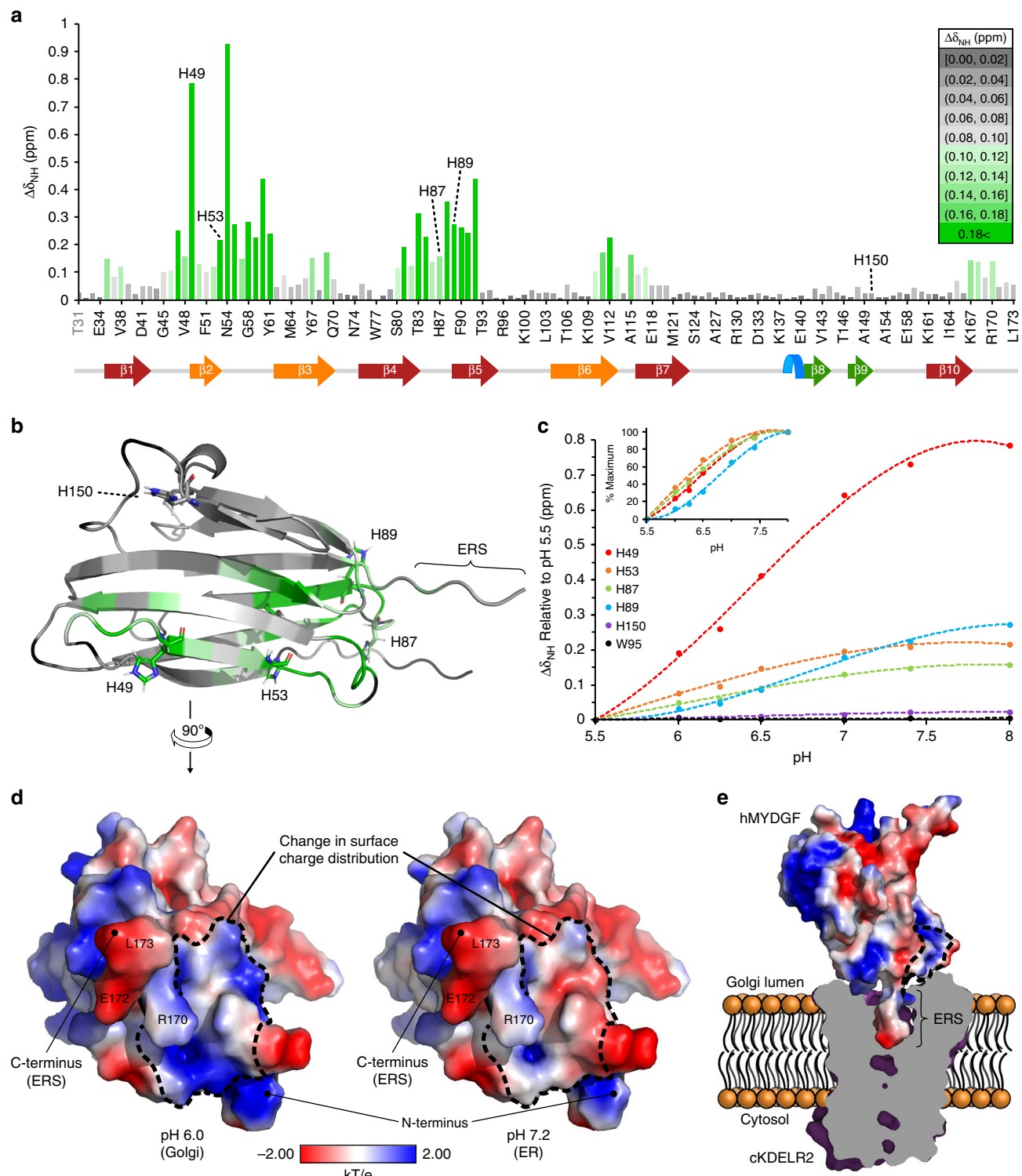

hMYDGF (Fig. 5a; residues from the cloning artifact are not shown). Among the most conserved residues are the two cysteines (Fig. 5a, center of the β-sandwich) and the two C-terminal residues that comprise the ERS. The ERS extends from an edge of the β-sandwich that contains predominantly variable residues (Fig. 5b). In contrast, the protein surface opposite the ERS, consisting of β1/β2, β3/β4, β5/β6, and β9/β10 loops, contains highly conserved residues (Fig. 5c). Underneath the surface of this conserved face between the β-sheets of the β-sandwich is an apparent cavity for which NOE interactions

between adjacent residue pairs lining the cavity were identified and validated, but no NOE interactions could be identified between residue pairs across it despite directed searches. When we analyzed the 20 most energetically stable conformers by Computed Atlas of Surface Topography of proteins (CASTp)[26] using the default probe radius of 1.4 Å (comparable to that of a water molecule), the solvent-accessible volume in this cavity ranged from 6 Å³ to 67 Å³, with an average of 37 Å³ and standard deviation of 16 Å³ (Fig. 5d; cavity in red for the most energetically stable structure). The side-chains identified by CASTp to line the

**Fig. 3 pH-dependent alterations of hMYDGF backbone $^{1}$H, $^{15}$N NMR peaks and surface charge distribution. a** Absolute value magnitude of the change in the combined $^{1}$H, $^{15}$N chemical shift of each residue between pH 5.5 and 8.0. The $\Delta\delta_{NH}$ values were binned and color-coded from lowest (gray) to highest (green) perturbation, with four of the five hMYDGF histidine residues falling into the highest bin. **b** Color-coded $\Delta\delta_{NH}$ bins from **a** mapped onto the structure of hMYDGF. Aside from H150, the hMYDGF histidines (stick representations) mark regions of highest pH-sensitive peak perturbation between pH 5.5 to pH 8.0. Residues that could not be assigned in at least one of the $^{1}$H, $^{15}$N HSQC spectra are colored black. **c** pH titration curves for the five hMYDGF histidines and W95 (no change) with $\Delta\delta_{NH}$ plotted for each pH condition relative to the pH 5.5 $^{1}$H, $^{15}$N HSQC spectrum. Curves for H49, H53, H87, and H89 are also graphed as a percent of maximum change (inset). Source data used for chemical shift perturbation calculations are provided as a Source Data file. **d** Surface charge distribution (blue, positive; red, negative; white, uncharged) of hMYDGF lacking the five N-terminal cloning residues calculated at pH 6 (lower end of the Golgi pH range) and pH 7.2 (pH in the ER) using PDB2PQR[18] and APBS based on $pK_a$ values predicted by PROPKA3.1 (refs. [19,20]). The most drastic changes were in a region enclosed by dotted lines in which the positive surface charge became more neutral/negative with increasing pH. This region was adjacent to the protruding ERS (three labeled residues). **e** hMYDGF depicting surface charge distribution at pH 6 docked onto cKDELR2 (PDB 6I6H (https://www.rcsb.org/structure/6I6H)[6]; purple) from HADDOCK cluster 1 (see Supplementary Fig. 4). cKDELR2 was clipped (interior cross-section in gray) to show how the hMYDGF ERS is situated into the cKDELR2 binding pocket. The region of greatest change in surface charge highlighted in **d** is largely at the interface of hMYDGF and cKDELR2 (enclosed by dotted lines) and may play a role in stabilizing the interaction in the Golgi and destabilizing for hMYDGF release in the ER.

cavity were well-conserved and exclusively hydrophobic (Fig. 5d; color-coded based on ConSurf conservation bin).

**MYDGF and vanin base domain protein homology.** The MYDGF family of proteins is categorized under UPF0556 (http://pfam.xfam.org/family/PF10572) in Pfam based on sequence homology. The hMYDGF NMR structure allowed us to search for structural similarity to other proteins using the Dali server, which matches and ranks the query structure to the structures of all the proteins in the PDB[27]. The server identified the base domain from the crystal structure of human vanin-1 (VNN1; pantetheinase; PDB 4CYF (https://www.rcsb.org/structure/4CYF)[28]) as the top match to the structure of hMYDGF, with a Dali z-score of 7.6 and 110 ordered residues of hMYDGF superimposing with a Cα RMSD of 4.0 Å (Fig. 6a, left; VNN1 in cyan). Vanins are a family of enzymes (comprising VNN1, VNN2, and VNN3 in humans) that hydrolyze a carboamide linkage in D-pantetheine, thus releasing cystamine and recycling pantothenic acid (vitamin B5)[29,30]. The base domain is linked to the N-terminal enzymatic, nitrilase domain[28]. VNN1 and VNN2 are ectoenzymes, which contain a base domain with a C-terminal glycosylphosphatidylinositol (GPI)-anchored cleavage site that is modified to tether the enzymes to the outer leaflet of the plasma membrane[29–31].

The human VNN1 base domain has the same β-strand connectivity as hMYDGF shown in Fig. 1b. The major structural differences are two additional disulfide bonds that are conserved in the vanin protein family (one between β2 and β3 strands and a second in the loop connecting β5 and β6 strands), hydrogen bonding between β6 and β9 to form a larger β-sheet (β2, β3, β6, β9, and β8) rather than the two separate sheets in hMYDGF, and a smaller pocket (10 Å$^3$) identified by CASTp in the vicinity of where the hMYDGF central cavity is located. A multiple sequence alignment of hMYDGF, human VNN1 base domain (UniProtKB: O95497 (https://www.uniprot.org/uniprot/O95497), residues V325-G491), and the Pfam seed sequences for MYDGF and vanin base domain (Pfam: PF10572 (http://pfam.xfam.org/family/PF10572) and Pfam: PF19018 (http://pfam.xfam.org/family/PF19018), respectively), revealed a number of conserved residues among the two protein families (Fig. 6b, consensus logo corresponds to the alignment from 82 sequences). Human MYDGF and VNN1 base domain have a sequence identity of 15%, with eight residues (asterisks) having ≥85% sequence identity among all sequences (black asterisks mark seven residues with ≥90% sequence identity). The eight residues are also conserved in over 75% of MYDGF homologs identified from the UniRef90 database (Fig. 5b). These residues appear to be structurally conserved as well, inasmuch as they are in close

proximity in the superposition of the two structures (Fig. 6a, right; VNN1 residues in cyan). The eight well-conserved residues include the disulfide connecting β3 to β5, five core residues that line the hMYDGF cavity (Fig. 5d), and a glycine located on the same face as the C-terminus.

## Discussion

Our goal in initiating these studies was to determine the solution structure of hMYDGF as a model for other members of the widespread MYDGF/UPF0556 family of proteins and to provide insights into the functions of MYDGF as a well-conserved resident ER protein[1] and as a paracrine/autocrine survival factor with therapeutic potential after myocardial infarction[7,32]. We found that hMYDGF consists of a β-sandwich occluded at one edge by an α-helical turn and small β-sheet (Fig. 1a, b). The protein is well-ordered except for the N-terminus, C-terminus (ERS), and elongated loop between β7 and the α-helix. A disulfide bond bridges the sheets of the β-sandwich, which, remarkably, encloses a cavity surrounded by hydrophobic residues (Fig. 5d). The structure allowed us to discern that the same fold is present in the "base domain" of the crystal structure of VNN1. Based on this discovery, a new Pfam entry was created for the vanin base domain called "Vanin_C" and given the accession PF19018 (http://pfam.xfam.org/family/PF19018). The similar fold and the presence of invariant residues are strong evidence that MYDGF and the vanin base are homologous protein domains.

The base domain of vanins tethers the nitrilase domain to the plasma membrane through a lipid anchor (Fig. 6c)[29]. As depicted in the same figure, ERS-bearing proteins such as MYDGF interact with multi-pass transmembrane KDELRs in the lumen of the Golgi[6]. The HADDOCK models of hMYDGF bound to cKDELR2 reveal a topology in which the ERS protrudes from the base of hMYDGF and the conserved loop residues are on the opposing face, available to interact with potential cargo. The nitrilase domain, in contrast, is at one side of the base domain in the crystal structure of VNN1 with a linker between the two domains wrapping around the base domain. KDELRs undergo a conformational change as a function of pH that favors binding of the ERS at pH 6 as found in the Golgi and release of the ERS at the higher pH found in the ER[4,6,13]. Although the global fold of hMYDGF is stable across this pH range, groups of residues, predominantly at the base of the β-sandwich from which the ERS protrudes and including three of the five hMYDGF histidines and surrounding residues, are sensitive to environmental pH as assessed by $^{1}$H, $^{15}$N HSQC perturbation analysis and surface charge distribution calculations. Docking hMYDGF onto cKDELR2 revealed that a number of these residues are at the

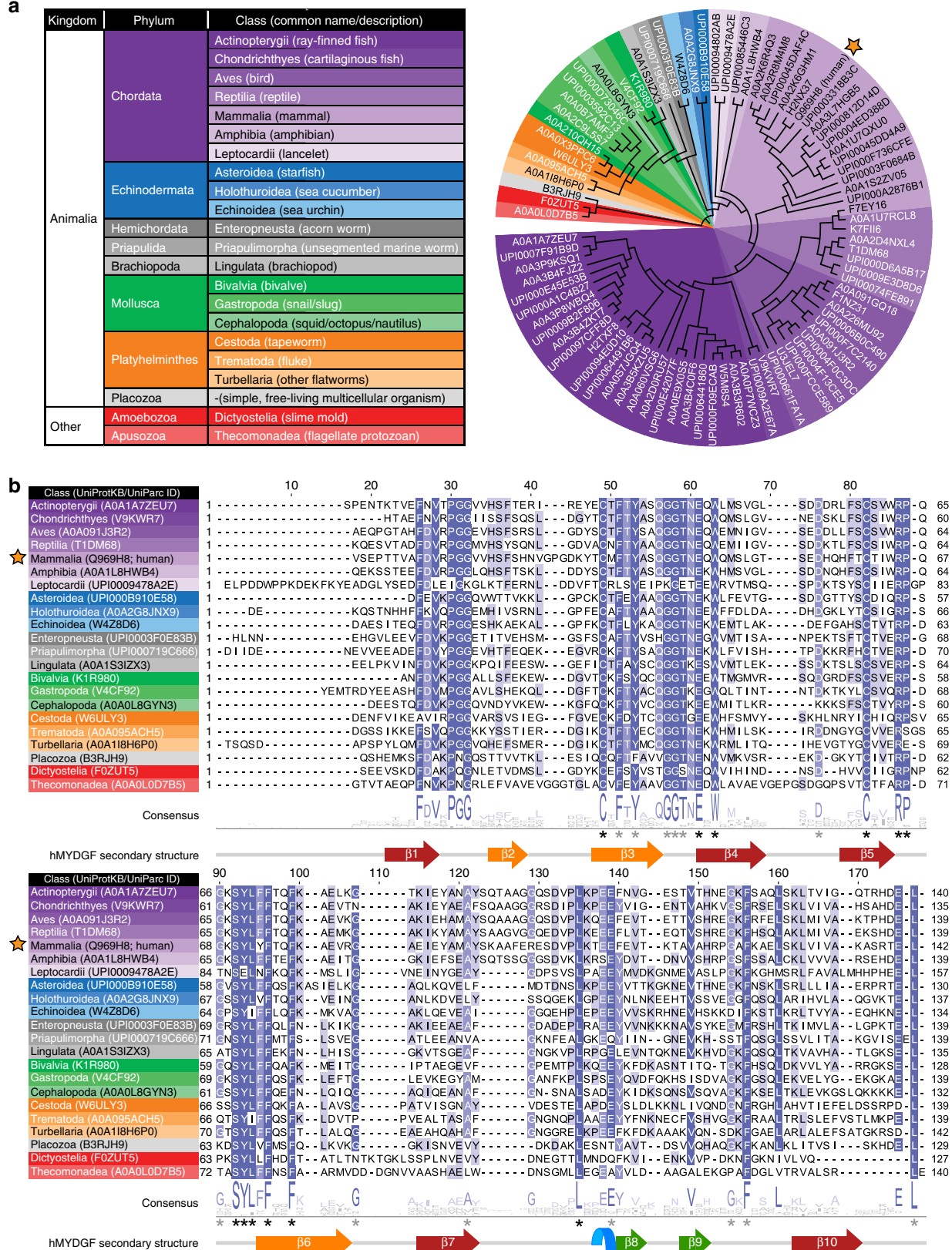

**Fig. 4 Homologs of hMYDGF. a** Phylogenetic tree color-coded by phylum and class of 87 MYDGF homologs identified by ConSurf in the UniRef90 database using the amino acid sequence of hMYDGF lacking its signal sequence (star) as the query. **b** Protein sequence alignment of MYDGF homologs from each of the classes in **a** with hMYDGF secondary structure displayed underneath. Asterisks represent highly conserved residues among all 236 UniProt90 MYDGF sequences identified by BLAST (black, ≥90% identity; gray, ≥85% identity).

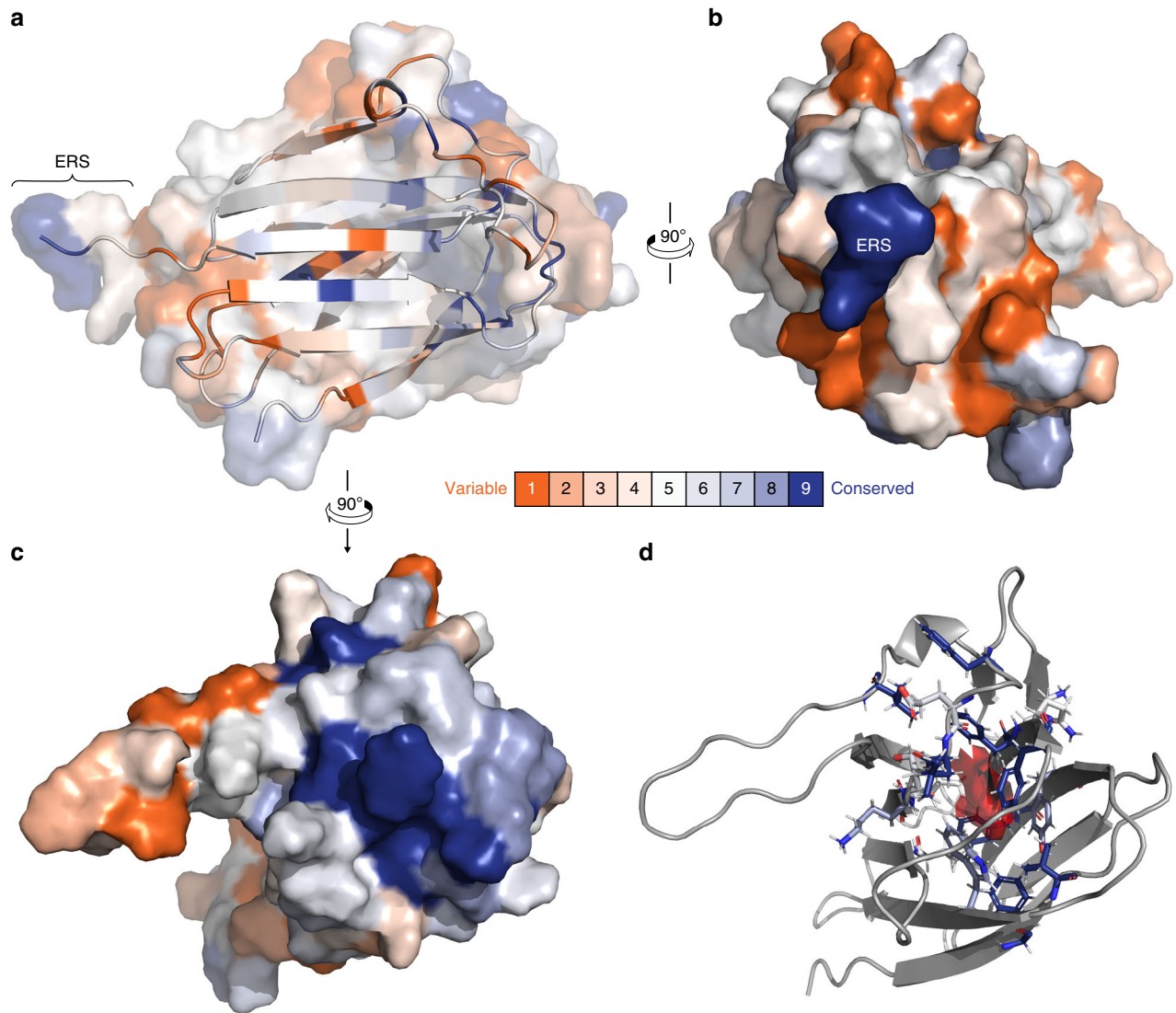

**Fig. 5 Amino acid conservation mapped onto the structure of hMYDGF. a** Using the ConSurf server, each residue of hMYDGF was binned from 1 (most variable, orange) to 9 (most conserved, blue) based on the alignment of 87 unique MYDGF protein homolog sequences (see Fig. 4) and mapped onto the structure of hMYDGF. In addition to the two cysteines that form the disulfide in the core of β-sandwich, the most conserved residues include, as highlighted in **b**, **c**, the C-terminus Glu-Leu sequence of the ERS and those in loops opposite the ERS. **d** A cavity (red) lined with conserved, hydrophobic residues (sticks; color-coded by conservation) beneath the surface shown in **c** was identified by CASTp and recreated in PyMOL as displayed here. The cavity has an average volume of 37 Å$^3$ among the 20 most energetically stable conformers that represent the structure of hMYDGF.

binding interface between the two proteins, raising the possibility that pH-dependent changes in both MYDGF and KDELRs modulate MYDGF–KDELR interactions. The significance of the two predicted modes of MYDGF binding to KDELR2 at pH 6 is not known.

MYDGF homologs are present in organisms throughout and beyond the animal kingdom, including species as distant to humans as slime molds and protozoans. Conservation analysis using ConSurf identified the C-terminal Glu-Leu residues that are part of the ERS among the most conserved residues (Fig. 5; bin 9, dark blue), indicating that the ERS is a critical component of nearly all MYDGF-family proteins. Of the most-conserved MYDGF residues in addition to those in the ERS, 12 do not overlap with conserved residues shared with vanin base family members (Fig. 6b, asterisks) and are highlighted as dark blue sticks in Fig. 6d. Eight of these residues reside in the loops on the opposite face from the ERS and the remaining four are between the β-sheets of the β-sandwich and surround the cavity (Fig. 6d,

red). While none of the 20 conformers individually has a clear inlet to the cavity, an overlay of these conformers reveals a potential entrance path from underneath the β9–β10 loop. We hypothesize that these conserved hMYDGF residues constitute a dynamic external (loops) and/or internal (cavity) binding interface for conserved and as-of-yet unknown interactor(s). Binding to this conserved interface of KDELR-tethered MYDGF is a potential means for the cell to return metabolic intermediates or non-ERS-containing proteins to the ER.

A possible alternative function of MYDGF is as part of the "tool kit" of ER components that assist in protein folding and movement through the ER. MYDGF itself likely does not require interaction with protein-folding machinery. The protein produced in *E. coli* renatured readily after exposure to 8 M urea into a structure with all peptidyl-prolyl peptide bonds in the trans configuration and a disulfide that is buried and apparently inert. There is no site for N-linked glycosylation in MYDGF, and we detected no post-translational modifications by MS. How

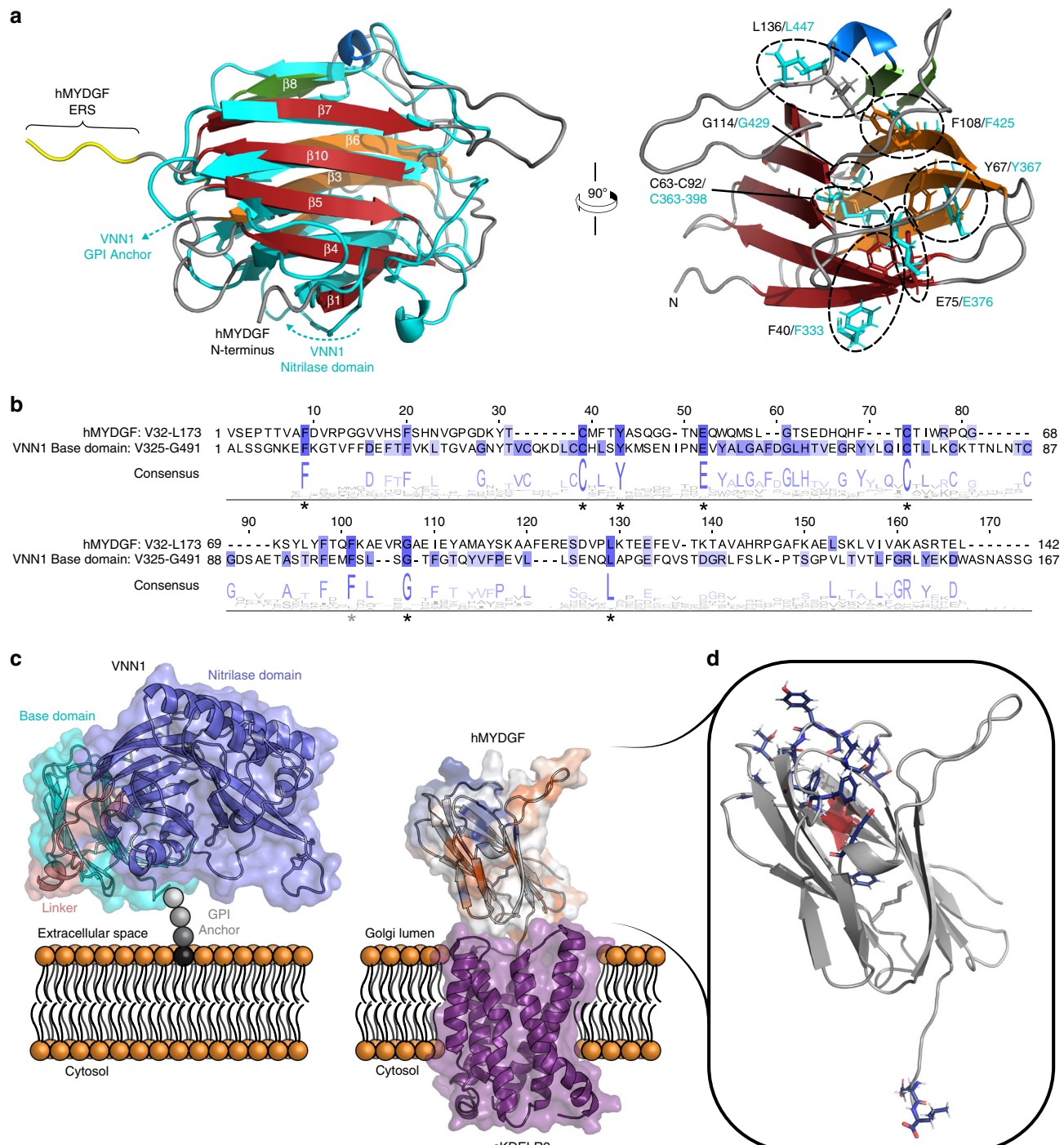

MYDGF, once folded, might function in the ER in not obvious. We note that the ECOD database, which contains a structural classification of proteins of known structure[33], groups the vanin base domain into a superfamily that includes the C-terminal domain of hyaluronate lyase and other related polysaccharide lyase enzymes. While these structural similarities are more distant than that found between MYDGF and the vanin base domain, the similarities suggest that MYDGF may interact with polysaccharides in ER/Golgi.

Our structure is useful for thinking about MYDGF as a paracrine/autocrine survival factor with therapeutic potential after cardiac ischemia[7,32]. A number of ER proteins other than MYDGF have activities outside of the cell, including GRP78 (BiP)[34], GRP94[35], and mesencephalic astrocyte-derived neurotrophic factor

(MANF)[36–38]. Yet to be determined are the receptor for the survival function of extracellular MYDGF and when such a function arose during evolution. Our analysis of the structure could aid in exploring these unknowns. We catalog residues that are conserved throughout the MYDGF family and may interact with an ancient receptor and conversely those that are more variable and may interact with a receptor that appeared more recently, e.g., along with hematopoietic and circulatory systems.

## Methods

**Expression and purification of recombinant hMYDGF.** Human blood eosinophil RNA and random primers were used to synthesize cDNA by reverse transcription polymerase chain reaction using the SuperScript III First-Strand Synthesis System (18080051, Thermo Fisher Scientific)[1]. *MYDGF* cDNA encoding the mature,

**Fig. 6 Comparison of hMYDGF and human VNN1 base domain. a** Left: superposition of ordered residues (P35-A126, D133-A168) from the NMR structure of hMYDGF upon the crystal structure of the human VNN1 base domain (PDB 4CYF (https://www.rcsb.org/structure/4CYF)[28]; cyan). 110 residues aligned with a Cα RMSD of 4.0 Å. The VNN1 base domain is flanked by a nitrilase domain at the N-terminus and a GPI anchor at the C-terminus (cyan arrows highlight the strands of the base domain that lead to these features). Right: stick representations of highly conserved residues (circled) shared by human VNN1 (cyan) and hMYDGF superimposed on the hMYDGF structure. **b** Multiple sequence alignment and consensus logo of hMYDGF, human VNN1 base domain, and Pfam seed sequences for MYDGF (Pfam: PF10572 (http://pfam.xfam.org/family/PF10572)) and vanin base domain (Pfam: PF19018 (http://pfam.xfam.org/family/PF19018)) families (82 sequences total). hMYDGF and the human VNN1 base domain are displayed as representative sequences, with hMYDGF sharing 15% sequence identity with the VNN1 base domain. Asterisks represent eight highly conserved residues among all sequences (black, ≥90% identity; gray, ≥85% identity), which align well in space between hMYDGF and the human VNN1 base domain as shown in **a**. **c** The VNN1 base domain (cyan) is connected to the nitrilase domain (blue) via a linker strand (pink) at its N-terminus and tethered to the plasma membrane by a GPI anchor at the C-terminus. An analogous model of hMYDGF (colored by residue conservation as in Fig. 5: blue, most conserved; orange, most variable) is presented in a similar orientation as the VNN1 base domain and engaging cKDELR2 (PDB 6I6H (https://www.rcsb.org/structure/6I6H)[6]; purple) from HADDOCK cluster 1 (Supplementary Fig. 4) in the Golgi via its C-terminal ERS with the remaining highly conserved residues on the opposite face. **d** The 14 most-conserved MYDGF residues (Fig. 5, bin 9) that do not overlap with conserved residues shared with vanin base family members (asterisks in **b**) are highlighted as dark blue sticks. Note that aside from the C-terminal Glu-Leu residues of the ERS, these MYDGF-specific conserved residues reside in the loops on the face opposite to the ERS and surface of the cavity (red).

human protein (V32-L173; UniProt Q969H8 (https://www.uniprot.org/uniprot/Q969H8)) was subsequently amplified by polymerase chain reaction and cloned into the pAcGP67.coco plasmid[39] using 5′ primer CCG GCG GAT CCG GTG TCC GAG CCC ACG ACG (BamHI restriction site) and 3′ primer GC TGC TTC TAG AAG CTC AGT GCG CGA TGC CTT GG (XbaI restriction site), and into the pET.ELMER plasmid[40] using 5′ primer CCG GCG GGT ACC GTG TCC GAG CCC ACG ACG GTG (KpnI restriction site) and 3′ primer CAG GGC GCT AGC TCA CAG CTC AGT GCG CGA TGC C (NheI restriction site).

Sf9 insect cells (11496015, Thermo Fisher Scientific) were co-transfected with the pAcGP67.coco-*MYDGF* plasmid and Sapphire Baculovirus DNA (ABP-BVD-10001, Allele Biotechnology), from which high titer stocks of recombinant baculovirus containing *MYDGF* were obtained by plaque purification and three rounds of amplification[39]. High Five insect cells (B85502, Thermo Fisher Scientific) were then infected with the recombinant baculovirus for hMYDGF production[39]. In this system, hMYDGF was expressed with an N-terminal gp67 signal sequence and a C-terminal 6xHis-tag that masks the ERS, ultimately resulting in the protein's secretion. The recombinant protein was purified from the medium by IMAC using nickel-nitrilotriacetic acid resin (30230, Qiagen). The purified insect cell-derived hMYDGF construct contained N-terminal residues ADP and C-terminal residues LELVPRGSAAGHHHHHH as a result of the cloning process.

hMYDGF was also expressed in and purified from either BL21(DE3) (69450, MilliporeSigma) or Rosetta 2(DE3) competent cells (71400, MilliporeSigma) transformed with the pET.ELMER-MYDGF plasmid. The only observable difference between the competent cells was that use of Rosetta 2(DE3) resulted in slightly higher protein yield. Transformed cells were grown at 37 °C in LB medium containing 30 μg/mL kanamycin and 25 μg/mL chloramphenicol until the OD$_{600}$ was between 1 to 1.5, upon which protein expression was induced with 1 mM isopropyl β-D-1-thiogalactopyranoside (IPTG) for 4 h. Cells were harvested by centrifugation then lysed by freeze–thaw and incubation in a cell lysis solution (100 mM sodium phosphate, 10 mM Tris Base, 8 M urea, 5 mM imidazole, 1 mM β-mercaptoethanol, pH 8). The recombinant protein, which was expressed with N-terminal residues MGGSHHHHHHGSLVPRGSKGT preceding the mature hMYDGF sequence, was purified by IMAC, dialyzed against dilute acetic acid (100 mM, then 1 mM; pH 3.7) to remove components of the cell lysis solution, dialyzed against phosphate-buffered saline (PBS; pH 6) or TBS (pH 8.5), and purified by size-exclusion chromatography on a HiLoad 16/600 Superdex 75 prep grade column (28-9893-33, GE Healthcare) equilibrated in and eluted with the same butter. In some batches, the N-terminal 6xHis-tag was cleaved off leaving only cloning residues GSKGT at the N-terminus (labeled as G27-T31 in figures). This was performed prior to size-exclusion chromatography by incubation for 4 h in a room temperature reaction in TBS using 0.01 units of thrombin (HT 1002a, Enzyme Research Laboratories) per 1 μg of recombinant hMYDGF. Bacteria-derived hMYDGF was concentrated after size-exclusion chromatography using Amicon Ultra-4 centrifugal filters (UFC801024, MilliporeSigma) and dialyzed against desired buffer depending on the experiment (specified for each section; all within ±0.02 pH units from reported pH). Protein concentrations were determined using Beer's law, with absorbance measurements at 280 nm obtained using a SpectraMax M5 Microplate Reader equipped with SoftMax Pro v6.3 software (Molecular Devices) and extinction coefficients predicted by the ExPASy ProtParam tool.

**Top-down MS**. Insect cell-derived hMYDGF (containing the C-terminal 6xHis-tag) and bacteria-derived hMYDGF (lacking the N-terminal 6xHis-tag) were dialyzed against 1 mM acetic acid (pH 3.7) and diluted with equal volume of acetonitrile to 12.5 μg/ml for top-down MS and MS/MS analysis. Both MS and MS/MS were performed on a Bruker maXis II ETD quadrupole time-of-flight mass spectrometer (Bruker Daltonics, Bremen, Germany) by direct infusion at a flow rate of

6 μL/min. Mass spectra of insect cell-derived hMYDGF and bacteria-derived hMYDGF were acquired at a scan rate of 1 Hz over 200–2000 *m/z* and 500–3000 *m/z* range, respectively. Targeted MS/MS of CID was performed with the selected charge states of insect cell-derived MYDGF (21+) at 1 Hz over 200–2000 *m/z*. The isolation window was set to 2 *m/z*, and the collision direct current bias was fixed to 23 V. Targeted MS/MS of CID and ETD was performed with the selected charge states of bacteria-derived MYDGF (19+) at 1 Hz over 500–3000 *m/z* range. The isolation window was set to 2 *m/z*. The collision direct current bias was fixed to 18 V for CID. The precursor ion accumulation was set to 800 ms with a reagent (3,4-hexanedione) injection duration of 8 ms and an additional 0 ms reaction for ETD. The MS spectra were deconvoluted by the Maximum Entropy algorithm with a resolving power of 80,000 using Bruker DataAnalysis 4.3 (Supplementary Fig. 1). All fragment ions from both CID and ETD were manually validated using MASH Suite Pro[41]. Fragments of $b$, $y$, $c - 1$, $z'$, and $z' + 1$ ions were assigned with a mass tolerance of 15 ppm after the validation to generate the ion fragment maps (Supplementary Fig. 1). All masses are reported as the monoisotopic masses.

**CD spectroscopy**. Far UV CD spectroscopy data were collected on an AVIV Model 420 Circular Dichroism Spectrometer and analyzed by Igor Pro v6.3 software. Scans were collected at 25 °C using the following parameters: 0.1 cm cuvette pathlength, 1 cm bandwidth, 1 nm steps, and 10 s averaging time. All scans were subtracted from corresponding buffer scans and converted to units of molar ellipticity. For comparison between 6xHis-tagged insect cell- and bacteria-derived MYDGF (Supplementary Fig. 1e), the insect cell-derived protein (5.9 μM) was dialyzed against 10 mM sodium phosphate, 10 mM NaCl (pH 7.5) and bacteria-derived protein (4.8 μM) was dialyzed against 10 mM sodium phosphate (pH 6). For the pH titration scans (Fig. 2a), bacteria-derived MYDGF lacking the 6xHis-tag (concentrations ranging 4.4–4.7 μM) was dialyzed in 10 mM sodium acetate for pH 4.0–5.5 conditions, and 10 mM sodium phosphate for pH 6.0–7.5 conditions. The BeStSel server[12] was used for secondary structure prediction based on the CD spectra (Supplementary Table 1) and the CD Analysis and Plotting Tool (CAPITO)[42] was utilized to smooth the CD spectra by applying a Savitzky–Golay filter for clearer visual comparison among the scans.

**Intrinsic tryptophan fluorescence**. Intrinsic tryptophan fluorescence of insect cell-derived hMYDGF (containing an N-terminal 6xHis-tag) and bacteria-derived hMYDGF (lacking the N-terminal 6xHis-tag) in TBS (pH 7.4) was measured using a Fluoromax-3 Spectrofluorometer in conjunction with Datamax v2.2 spectroscopy software (HORIBA Jobin Yvon). When comparing insect cell- and bacteria-derived hMYDGF (Supplementary Fig. 1f), scans were collected in 2 mm × 10 mm quartz cuvettes at 25 °C using the following parameters: excitation λ: 295 nm, emission λ: 305–400 nm, excitation bandwidth: 2 nm, emission bandwidth: 4 nm, 0.5 nm steps, and 0.5 s integration time. The same conditions/parameters were used for analysis of native versus denatured and/or reduced bacteria-derived hMYDGF (Supplementary Fig. 3b), with the exception of the following: excitation bandwidth: 2 nm, emission bandwidth: 4 nm, and 1 s integration time. All samples contained 1 μM recombinant hMYDGF. Resulting spectra were an average of three scans that were baseline subtracted from their corresponding, buffer scans. Fluorescence peak maximums were determined from sixth-order polynomial trendlines, which were fit to the spectra using Microsoft Excel.

**Production of isotopically labeled MYDGF**. The method for producing single ($^{15}$N) and double ($^{13}$C, $^{15}$N) labeled hMYDGF was adapted from multiple protocols[43–45]. A 1-mL glycerol stock of Rosetta 2(DE3) cells transformed with pET.ELMER-MYDGF in MDG medium[43] was used to inoculate 50 mL of sterile MDG

medium containing 60 µg/mL kanamycin and 50 µg/mL chloramphenicol. Culture was grown overnight at 25 °C in a shaking incubator. The following day, 1 L of sterile-filtered, heavy isotope-enriched minimal growth medium was prepared composed of the following: 1× M9 salts[44], 0.1× metal mix[43], 2 mM MgSO$_4$, 0.4% w/v glucose ($^{13}$C-glucose for double-labeled sample), 0.2% w/v $^{15}$NHCl$_4$, 0.1 mM CaCl$_2$, 1× vitamin solution[45], 1× vitamin B$_{12}$ solution[45], 30 µg/mL thiamine, 60 µg/mL kanamycin, and 50 µg/mL chloramphenicol. One liter of minimal growth medium was inoculated with 20 mL of overnight culture and incubated at 37 °C with shaking until OD$_{600}$ ≈ 1. Protein expression was induced with 1 mM IPTG for 25 h at 25 °C with shaking. The protocol described above was performed to extract, renature, remove the 6xHis-tag, purify from the cleavage mix, dialyze against desired buffer, and determine concentration for the isotopically labeled proteins. NMR samples were supplemented to achieve final concentrations of the following: 10% v/v D$_2$O, 15 µM 4,4-dimethyl-4-silapentane-1-sulfonic acid (internal chemical shift reference), and 0.02% w/v NaN$_3$ (bacteriostat).

**NMR data collection and processing**. NMR spectra were acquired on Bruker AVANCE III or Varian VNMRS spectrometers ranging from 600 MHz to 900 MHz using TopSpin 3.5 and VNMRJ, respectively, and equipped with cryogenic triple-resonance probes. The temperature of the sample was regulated at 298 K for all recorded experiments. Data processing was conducted using the NMRPipe package[46]. Non-uniform sampling (NUS) was employed for the triple resonance experiments with sampling rates ranging from 30 to 40%. A Poisson-gap sampling[47] schedule was used for the NUS data acquisition and the SMILE plug-in[48] in NMRPipe was used for spectral reconstruction.

**NMR structure determination of MYDGF**. Backbone resonances were manually assigned in NMRFAM-SPARKY[49] using 2D $^1$H, $^{15}$N HSQC, 3D CBCA(CO)NH, 3D HNCACB, and 3D HNCO spectra with an additional 3D $^1$H, $^{15}$N HSQC - $^1$H, $^{13}$C HSQC NOESY experiment used for assigning the A157 backbone amide (4.6 ppm in the $^1$H-dimension). All backbone amides were assigned aside from the first two cloning residues of the recombinant protein (labeled as G27–G28 in figures). Side chain resonances were also manually assigned using 2D $^1$H, $^{13}$C HSQC (aliphatic), 2D $^1$H, $^{13}$C HSQC (aromatic), 3D C(CO)NH, 3D HBHA(CO)NH, 3D H (CCO)NH, 3D HCCH-TOCSY (aliphatic), 3D HCCH-TOCSY (aromatic), 2D (HB)CB(CGCD)HD (aromatic), and 2D (HB)CB(CGCDCE)HDHE (aromatic) spectra. Manually assigned resonances were cross-validated using the I-PINE web server[50] after submitting a job with the PINE-SPARKY.2 plugin[51], and the structure-independent probabilistic validation algorithm ARECA[52]. The $^{13}$C chemical shifts for the Cα and Cβ of the two hMYDGF cysteines (C63, C92) also matched reported resonances of cysteines that form disulfides in β-sheets of other proteins[53].

The solution structure of $^{13}$C, $^{15}$N hMYDGF (508 µM) in PBS (pH 6) was solved using an Integrative NMR approach[54] with three additional NMR spectra: 3D $^1$H, $^{15}$N HSQC NOESY, 3D $^1$H, $^{13}$C HSQC NOESY (aliphatic), and 3D $^1$H, $^{13}$C HSQC NOESY (aromatic). Initial folding was calculated with the PONDEROSA refinement option, which utilizes CYANA[55] for obtaining inter-residual proximity information used in the Ponderosa Analyzer white-list/black-list manager for efficient NOE assignment in consequent steps. Xplor-NIH-based calculations (AUDANA[56]) were used for the remaining majority of submissions in the PONDEROSA-C/S package[57]. Distance and angle constraints were generated and used for structure calculation by running PONDEROSA-X refinement, which utilized the AUDANA algorithm and TALOS-N[58] optimization from the resonance assignment list and NOESY spectra inputs. Constraints were carefully validated using Ponderosa Analyzer interfaced with the Ponderosa Connector plug-in (two-letter-code "up") in NMRFAM-SPARKY and PyMOL 2.0 programs. After several iterative calculations using the Constraints Only-X option, we finalized constraint refinement by running Final Step with the explicit water refinement option. This provided the top 20 out of 100 lowest pseudo-potential energy conformers and conducted energy minimization in the water box. The final structures were validated using the wwPDB Validation Service (https://validate-rcsb-2.wwpdb.org/)[15,16] and MolProbity[59] online servers. Structural NMR statistics for the 20 most energetically stable hMYDGF conformers are summarized in Table 1.

**MYDGF relaxation times and hydrogen exchange with solvent**. $^{15}$N relaxation experiments ($T_1$, $T_2$, and heteronuclear NOE; Supplementary Fig. 2), and amide hydrogen exchange spectra (Supplementary Fig. 3c, d) were recorded on Varian VNMRS spectrometers operating at 600 and 800 MHz, and equipped with a cryogenic triple-resonance probe. The same sample that was used for structure determination was also used for collecting these NMR experiments. All spectra were recorded with the temperature of the sample regulated at 298 K.

For measuring $^{15}$N $T_2$ values, multiple 2D $^1$H,$^{15}$N spectra were recorded in an interleaved fashion using relaxation delays of 10, 30, 50, 70, 90, 110, 130, 150, 170,190, and 210 ms. Similarly, $^{15}$N $T_1$ values were measured using relaxation delays of 80, 160, 240, 320, 400, 560, 720, 960,1200, 1520, and 2000 ms. $^{15}$N heteronuclear NOE experiments were recorded in duplicate using a relaxation delay of 5 s with and without saturation of the amide protons. All the spectra were processed in NMRPipe and analyzed in NMRFAM-SPARKY. $^{15}$N $T_1$ and $T_2$

relaxation times were calculated by fitting the decaying signals to a single exponential function using the "rh" extension in NMRFAM-SPARKY. $^{15}$N heteronuclear NOE values were calculated from the ratio of corresponding intensities between the spectra recorded with and without $^1$H saturation using the "np" extension.

To map SEA groups, amide hydrogen exchange experiments were acquired by using a clean SEA HSQC experiment[60]. In short, magnetization from all protein hydrogens is first eliminated by a double $^{15}$N/$^{13}$C filter and then allowed to recover through exchange with bulk water during a mixing period (ranging from 10 to 140 ms), such that only amides that are exposed to the solvent are observed. All spectra were processed with NMRPipe and analyzed in NMRFAM-SPARKY.

**Calcium and pH titration**. 2D $^1$H, $^{15}$N HSQC spectra of $^{15}$N hMYDGF (257 µM) in TBS (pH 7.4) were collected in the presence of 0.5-, 1-, 2-, and 4-fold molar excess of CaCl$_2$. The spectrum of the sample with the highest calcium concentration is compared to the spectrum of hMYDGF in the absence of calcium in Supplementary Fig. 5.

For pH titration, 2D $^1$H, $^{15}$N HSQC experiments were recorded at various pH values (Figs. 2b and 3a–c). $^{15}$N hMYDGF (concentrations ranging 215–356 µM) was dialyzed against 10 mM acetic acid, 10 mM sodium phosphate, 150 mM NaCl at pH 4.0, 5.5, 6.0, 6.25, 6.5, 7.0, and 8.0. hMYDGF at pH 4.0 precipitated out of solution and a suitable $^1$H, $^{15}$N HSQC spectrum could not be obtained. The same spectrum from the calcium titration studies (before addition of CaCl$_2$) was used as the pH 7.4 condition. All spectra were aligned to the W77 backbone amide peak which displayed one of the smallest chemical shift changes as a function of pH. The chemical shift perturbation ($\Delta\delta_{NH}$ in ppm) for each assigned peak was calculated using following equation:

$$\Delta\delta_{NH} = \sqrt{(\Delta\delta_H)^2 + (\Delta\delta_N/5)^2}, \qquad (1)$$

where $\Delta\delta_H$ and $\Delta\delta_N$ are the peak displacements (ppm) for the hydrogen and nitrogen dimensions, respectively. Peaks with the lowest displacement were categorized in the first bin of Fig. 3a (darkest gray). The upper bound of the first bin was set to the third quartile +1.5 × (interquartile range) of $\Delta\delta_{NH}$ values between the pH 6 $^1$H, $^{15}$N HSQC spectrum for this set of experiments and the pH 6 $^1$H, $^{15}$N HSQC spectrum of hMYDGF in PBS used for backbone resonance assignments, which had minimal overall peak perturbations. Consequent bins were set and color-coded based on multiples of the first, with the tenth bin (darkest green) containing the peaks with the highest $\Delta\delta_{NH}$ values. Trendlines fit to the $\Delta\delta_{NH}$ values of the five hMYDGF histidines and W95 (Fig. 3c) were third-order polynomials, with $pK_a$ values calculated from their second derivatives set to 0.

The surface charge distribution for the most energetically favorable NMR conformer of hMYDGF (lacking the five N-terminal cloning residues) was calculated at pH 6 and pH 7.2 (Fig. 3d, e) through the PDB2PQR server[18] using PROPKA3.1 (refs. [19,20]) to assign protonation states at the desired pH values. The APBS Tools 2.1 Plugin in PyMOL was then utilized to visualize the surface charge distribution.

**MYDGF docking onto KDELR2**. Predicted complexes between the most energetically stable NMR conformer of hMYDGF (omitting the five N-terminal residues introduced from the cloning process) and the crystal structure of cKDELR2 (PDB 6I6H (https://www.rcsb.org/structure/6I6H)[6]) were generated using the webserver implementation of the computational docking program HADDOCK[21–23]. This program is predicated on the fact that knowledge of the contacting interfacial residues in experimentally determined complexes involving homologous proteins can be used as soft restraints when predicting a given unknown complex. The experimentally solved complex between cKDELR2 and the ERS-containing peptide TAEKDEL at pH 6 provided information regarding the cKDELR2 residues (specifically R5, R47, Y48, E117, R159, Y162, N165, W166, and R169) that are implicated in binding ERS sequences in the Golgi by forming a network of hydrogen bonds[6]. The soft restraints that the HADDOCK webserver uses in its docking calculation were specified by these cKDELR2 residues in addition to the hMYDGF ERS sequence (residues RTEL). In brief, the algorithm consists of three steps: (1) randomization of the orientation of one of the proteins and rigid body energy minimization, (2) simulated annealing and energy minimization in torsion angle space, and (3) refinement with energy minimization and molecular dynamics in explicit water, retaining soft pairwise restraints between the constituent atoms of the user-specified residues at all three stages (for additional algorithmic details, refer to cited publications[21–23]). HADDOCK grouped 400 predicted output complexes into two relatively equal-sized clusters (statistics summarized in Supplementary Table 2). These clusters had average energetic scores (unitless HADDOCK scores) of −94.9 ± 3.6 and −73.2 ± 0.6, which are calculated as weighted sums of various energies and buried surface areas (the cluster with the lowest HADDOCK score is ranked first). The lowest-energy complexes from each of these two clusters were examined for further analysis (Supplementary Fig. 4).

**Analysis of MYDGF sequence similarity**. MYDGF sequence similarity was mapped onto the structure of hMYDGF using the ConSurf web server[25]. A PDB file of hMYDGF (lowest pseudo-potential energy structure lacking the first five cloning residues) was submitted to ConSurf and a multiple sequence alignment was

generated with parameters set to use the HMMER homolog search algorithm (1 iteration, 0.0001 E-value cutoff) against the UniRef90 protein database. Homolog selection parameters were set to 500 sequences that sample the list of homologs to the reference sequence, with cutoffs of 100% for maximum identity between sequences and 20% for minimum identity for homologs. MAFFT(L-INS-i) was used to build the multiple sequence alignment of the 87 identified, unique sequences and the Bayesian calculation method was used to determine the rate of evolution at each site in the alignment. Resulting ConSurf output files were used as the basis for creating a phylogenetic tree using the Interactive Tree Of Life (iTOL v4) web server[61] and mapping sequence homology onto the structure of hMYDGF in PyMOL (Figs. 4a and 5).

Jalview v2.1 was used to generate, visualize, and create images of protein sequence alignments. A multiple sequence alignment that samples the different classes of MYDGF homologs was generated by MAFFT(L-INS-i) using protein sequences derived from UniProt (Fig. 4b; UniProtKB/UniParc IDs listed next to class). Signal peptides, either annotated in UniProt or predicted using the SignalP 5.0 server[62], were omitted from this alignment. If a signal peptide could not be identified, the C-terminal truncated protein sequence identified by ConSurf was used instead. A MAFFT(L-INS-i) alignment of 236 representative UniRef90 sequences found through BLAST (mature hMYDGF input sequence, 0.0001 E-value cutoff) identified 25 residues with ≥85% sequence identity across all sequences (Fig. 4b, asterisks; black asterisks correspond to 12 residues with ≥90% homology).

**MYDGF and vanin base sequence and structural homology**. The Dali server[27] was used to identify proteins structurally similar to the hMYDGF structure. The most energetically stable conformer lacking the first five cloning residues (mature hMYDGF) was used as the query protein structure against the full PDB. The crystal structure of the human VNN1 base domain (PDB 4CYF (https://www.rcsb.org/structure/4CYF[28]), residues V314–G491) was identified as the top hit with a Dali Z-score of 7.6. All other hits had a Dali Z-score of <5 (scores <2 are considered invalid hits) and none appeared to have a complete hMYDGF fold. A total of 110 hMYDGF ordered residues (P35-A126, D133-A168) superimposed with the human VNN1 base domain with a Cα RMSD of 4.0 Å using the default settings for the "super" command in PyMOL.

The MUSCLE alignment of 82 sequences for MYDGF- and vanin-family proteins (Fig. 6b) was generated through Jalview using their corresponding seed sequences from Pfam (Pfam: PF10572 (http://pfam.xfam.org/family/PF10572) and Pfam: PF19018, (http://pfam.xfam.org/family/PF19018), respectively) in addition to the mature hMYDGF sequence (UniProtKB: Q969H8 (https://www.uniprot.org/uniprot/Q969H8), residues V32-L173) and the human VNN1 base domain (UniProtKB: O95497 (https://www.uniprot.org/uniprot/O95497), residues V325-G491). This alignment was the basis for defining the human VNN1 base domain as residues V325-G491. Inasmuch as UniProtKB annotating the enzymatic domain as residues A39-S306, we define residues H307-E324 as a linker that connects the two domains of VNN1 (Fig. 6c).

**Reporting summary**. Further information on research design is available in the Nature Research Reporting Summary linked to this article.

## Data availability

The solution structure of hMYDGF along with structural restraints was deposited in the PDB under accession code 6O6W (https://www.rcsb.org/structure/6O6W). NMR data were deposited in BMRB under accession number 30584 (http://www.bmrb.wisc.edu/data_library/summary/?bmrbId=30584). MYDGF and vanin base protein families can be viewed in Pfam under accession codes PF10572 (http://pfam.xfam.org/family/PF10572) and PF19018 (http://pfam.xfam.org/family/PF19018), respectively. The lowest-energy hMYDGF/cKDELR2 complexes from HADDOCK clusters 1 and 2 were deposited in PDB-Dev under accession code PDBDEV_00000036 (https://pdb-dev.wwpdb.org/). The source data underlying Figs. 2a, 3a, c, Supplementary Table 1, and Supplementary Figs. 1e, f, and 3b are provided as a Source Data file. The data that support the findings of this study are available from the corresponding author upon reasonable request.

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

## Acknowledgements

We thank and acknowledge the support of William Milo Westler at NMRFAM for his input on the hMYDGF NMR structure; Ronnie Frederick for offering his protocols, expertise, and reagents for producing isotopically labeled protein; and Darrell McCaslin for offering advice on conducting CD experiments and interpreting results. V.B., D.S.A., and D.F.M. were supported by NIH grants P01 HL088594 and R01 AI125390. M.T., W.L., and J.L.M. were supported by NIH grant P41 GM103399. This study made use of the National Magnetic Resonance Facility at Madison, which is supported by NIH grant P41 GM103399, formerly P41 RR002301. Equipment was purchased with funds from the University of Wisconsin-Madison, the NIH (P41 GM103399, S10 RR002781, S10 RR008438, S10 RR023438, S10 RR025062, S10 RR029220), and the NSF (DMB-8415048, OIA-9977486, BIR-9214394). Z.L. and Y.G. acknowledge top-down proteomics software grant R01 GM125085 and high-end instrument grant S10 OD018475.

## Author contributions

D.S.A. cloned the human *MYDGF* gene into expression vectors and expressed and purified insect cell-derived hMYDGF. V.B. expressed and purified non-labeled and isotopically labeled, bacteria-derived hMYDGF, conducted C.D. and intrinsic tryptophan experiments, quantified resonance perturbation results, and generated phylogenetic trees. A.B. carried out sequence analysis and updated Pfam based on finding regarding MYDGF and vanin base homology. V.B. and A.B. generated multiple sequence alignments. M.T. collected and processed all NMR data, and performed relaxation experiment calculations; V.B. assigned and validated NMR resonances and analyzed N.O.E. constraints; V.B. and W.L. conducted structure calculation and validation. M.T., W.L., V.B., and J.L.M. assessed and validated NMR results. Z.L. collected and processed mass spectrometry experiments; V.B., Z.L., and Y.G. analyzed and validated mass spectrometry results. J.C.M. and O.N.D. carried out computational docking of hMYDGF onto cKDELR2. V.B. and D.F.M. planned the experiments and wrote the manuscript. All authors reviewed and edited the manuscript.

## Competing interests

The authors declare no competing interests.
