## [Peer Review File · Nature Communications]

Reviewers' comments:

Reviewer #1 (Remarks to the Author):

This study provides a comprehensive structural analysis of MYDGF, including calculation of the three dimensional structure, structural dynamics and hydrogen exchange. In addition, regions of the structure characterized by high sequence conservation were identified. Given this is the first structure for this family, which is widespread, the study is likely to have wide reaching implications. The structural similarity with human vanin-1 also expands the novelty of the study and has implications for the function and mechanism of action. Thought provoking ideas regarding the biology of MYDGF are presented at the end of the manuscript.

Sufficient detail appears to be given to repeat the study. In terms of confirming the two expressed proteins had the same fold, was it possible to compare the HSQC spectra of unlabeled material, or to use chromatographic methods?

The authors refer to the structure being 50% unstructured. Is this referring to a lack of regular secondary structure or disorder? The overlay of the structures implies the majority of the molecule is structured.

The CD spectra suggest that the overall fold is not changing, but does an analysis of the alpha carbons/protons support the conclusion that the overall fold is not changing?

Reviewer #2 (Remarks to the Author):

Myeloid-derived growth factor (MYDGF) has been recently discovered to mediate cardiac repair after myocardial infarction. This manuscript describes the work that led to obtaining the high-resolution structure for this factor. The authors outline a region on the protein surface as a putative binding interface for a MYDGF cargo protein based solely on the structural similarity to a domain from a member of the vanin family of proteins.

Unfortunately, the authors did not try to corroborate the results of their structural analysis with any biochemical/biological experiments that might shed light on the molecular functions of this factor. The most obvious would be testing the importance of the residues that are affected by pH for the MYDGF-KDEL interactions or exploiting the structural information in search for interacting proteins or probing the interaction with polysaccharides in ER.

In its present form, the work is just descriptive, lacking strong points and therefore only of a limited impact. It will not be of interest to the expert community outside of the field.

Minor comments:

The signal-to-noise ratio in the CD is extremely low, which prevents their interpretation.

Although, I understand the motivation for the pH titration experiments, there is no justification for calcium binding experiments.

Reviewer #3 (Remarks to the Author):

The manuscript by Bortnov et al. describes the solution NMR structural studies of human MYDGF. The work describes a high quality, novel NMR structure, pH and calcium titration studies to observe structural changes and a phylogenetic analysis which identifies human vanin-1 (VNN1) as having a similar fold. Interestingly, the authors compare VNN1's association to the "plasma membrane via a lipid anchor" with huMYDGF's C-terminal association to KDELs. Before I

recommend this for publication, I would like the authors to address the following questions/points:

-In Figure 1, there is an electrostatic surface representation that is included. However, it would be helpful to show how the surface charge distribution would change with different pHs, especially in the context of the C-terminus (where it is proposed to bind KDELR) and its surrounding residues. This would complement the NMR pH studies.

-For Figure 6c, were docking studies performed to generate the model of KDELR2 and huMYDGF? If so, how was the docking performed? If not, I would suggest using the recent KDELR structural studies (Bräuer et al., 2019) as a guide to build a docked model. In addition to the C-terminal ERS binding to the polar cavity of KDELR, there may be additional protein complementarity between KDELR and huMYDGF.

-The Bräuer paper should be referenced if PDB code 616H is mentioned in line 882.

-The authors mention a "dynamic binding interface" for potential interactors in line 315. This region should be labeled in the figure if possible. Was there any mutational work that supports this interface hypothesis? Is this interface region proximal to where VNN1 binds to the VNN1 nitrolase domain?

Reviewer #1

This study provides a comprehensive structural analysis of MYDGF, including calculation of the three dimensional structure, structural dynamics and hydrogen exchange. In addition, regions of the structure characterized by high sequence conservation were identified. Given this is the first structure for this family, which is widespread, the study is likely to have wide reaching implications. The structural similarity with human vanin-1 also expands the novelty of the study and has implications for the function and mechanism of action. Thought provoking ideas regarding the biology of MYDGF are presented at the end of the manuscript.

Sufficient detail appears to be given to repeat the study. In terms of confirming the two expressed proteins had the same fold, was it possible to compare the HSQC spectra of unlabeled material, or to use chromatographic methods?

- In order to compare the recombinant hMYDGF proteins purified from a baculovirus versus bacterial system by HSQC, hMYDGF from the baculovirus system would have to be isotopically labeled as well, which would require an extensive amount of time and resources. We were not able to obtain chromatographic data, as the protein yield from the baculovirus system was limited and our efforts focused on the use of top-down mass spectrometry, circular dichroism spectroscopy, and intrinsic tryptophan fluorescence to support the claim that both protein are in the same fold.

The authors refer to the structure being 50% unstructured. Is this referring to a lack of regular secondary structure or disorder? The overlay of the structures implies the majority of the molecule is structured.

- “Unstructured” in the text referred to lack of regular secondary structure and has been changed to “irregular” in the manuscript.

The CD spectra suggest that the overall fold is not changing, but does an analysis of the alpha carbons/protons support the conclusion that the overall fold is not changing?

- Although we have not assessed backbone $C\alpha/H\alpha$ as a function of pH, the chemical shifts of backbone amide resonances acquired by 1H , ^{15}N HSQC are sensitive to local environmental changes. Finding that the majority of the hMYDGF amide resonances remain in the exact same position across a range of pH indicates that the overall structure is unaltered.

Reviewer #2

Myeloid-derived growth factor (MYDGF) has been recently discovered to mediate cardiac repair after myocardial infarction. This manuscript describes the work that led to obtaining the high-resolution structure for this factor. The authors outline a region on the protein surface as a putative binding interface for a MYDGF cargo protein based solely on the structural similarity to a domain from a member of the vanin family of proteins.

Unfortunately, the authors did not try to corroborate the results of their structural analysis with any biochemical/biological experiments that might shed light on the molecular functions of this factor. The most obvious would be testing the importance of the residues that are affected by pH for the MYDGF-KDEL interactions or exploiting the structural information in search for interacting proteins or probing the interaction with polysaccharides in ER.

- We agree that identification of interacting partner(s) is needed to complete the story. We are in the midst of such experimentation and we are not certain how quickly we can make conclusions.

In its present form, the work is just descriptive, lacking strong points and therefore only of a limited impact. It will not be of interest to the expert community outside of the field.

Minor comments:

The signal-to-noise ratio in the CD is extremely low, which prevents their interpretation.

- In order to provide a clearer comparison of the overlaid CD spectra, CAPITO was used to smooth the CD spectra which are now included in the figures. We also now include a table that summarizes peak maxima, peak minima, and predicted secondary structure from multiple, unsmoothed hMYDGF CD spectra. Source data for all spectra are provided as a Source Data file as well.

Although, I understand the motivation for the pH titration experiments, there is no justification for calcium binding experiments.

- With cellular ER calcium concentrations reaching into the millimolar range and MYDGF being an ER-resident protein, we wanted to determine whether the structure was impacted by presence of calcium. In addition, the Trychta *et al. Cell Rep.* 2018 paper describes that ER calcium depletion results in ER stress and secretion of hMYDGF. This justification is now explained in the manuscript.

Reviewer #3

The manuscript by Bortnov et al. describes the solution NMR structural studies of human MYDGF. The work describes a high quality, novel NMR structure, pH and calcium titration studies to observe structural changes and a phylogenetic analysis which identifies human vanin-1 (VNN1) as having a similar fold. Interestingly, the authors compare VNN1's association to the "plasma membrane via a lipid anchor" with huMYDGF's C-terminal association to KDELRs. Before I recommend this for publication, I would like the authors to address the following questions/points:

In Figure 1, there is an electrostatic surface representation that is included. However, it would be helpful to show how the surface charge distribution would change with different pHs, especially in the context of the C-terminus (where it is proposed to bind KDELR) and its surrounding residues. This would complement the NMR pH studies.

- The surface charge distribution at pH 6 (representing pH in the Golgi) and pH 7.2 (representing pH in the ER) was calculated using PDB2PQR and APBS, which is now presented in Figure 3 and elaborated on in the text.

For Figure 6c, were docking studies performed to generate the model of KDELR2 and huMYDGF? If so, how was the docking performed? If not, I would suggest using the recent KDELR structural studies (Bräuer et al., 2019) as a guide to build a docked model. In addition to the C-terminal ERS binding to the polar cavity of KDELR, there may be additional protein complementarity between KDELR and huMYDGF.

- Formal docking analysis using HADDOCK is now included in Figure 3e, 6c, Supplementary Table 2, and Supplementary Figure 4. Docking was accomplished in collaboration with Dr. Julie Mitchell and Dr. Omar Damerdash, who have been added as co-authors with the explicit permission from each of the original authors. Evidence for the hypothesized complementarity is now presented.

The Bräuer paper should be referenced if PDB code 616H is mentioned in line 882.

- Reference has been added to the manuscript.

-The authors mention a "dynamic binding interface" for potential interactors in line 315. This region should be labeled in the figure if possible. Was there any mutational work that supports this interface hypothesis? Is this interface region proximal to where VNN1 binds to the VNN1 nitrolase domain?

- The residues involved in the putative binding interface are now better highlighted in Figure 6d. There was no mutational work to support this interface, but this will be accomplished once we have identified candidate interactors. The VNN1 enzymatic domain is situated adjacent to this corresponding interface of the base domain. The VNN1 base domain is now shown in a similar orientation to hMYDGF in Figure 6c for easier comparison.